# Diagnostic Accuracy of Machine Learning Models on Mammography in Breast Cancer Classification: A Meta-Analysis

**DOI:** 10.3390/diagnostics12071643

**Published:** 2022-07-05

**Authors:** Tengku Muhammad Hanis, Md Asiful Islam, Kamarul Imran Musa

**Affiliations:** 1Department of Community Medicine, School of Medical Sciences, Universiti Sains Malaysia, Kubang Kerian 16150, Kelantan, Malaysia; tengkuhanismokhtar@gmail.com; 2Department of Haematology, School of Medical Sciences, Universiti Sains Malaysia, Kubang Kerian 16150, Kelantan, Malaysia; 3Institute of Metabolism and Systems Research, University of Birmingham, Birmingham B15 2TT, UK

**Keywords:** machine learning, diagnostic accuracy, mammography, meta-analysis, breast cancer

## Abstract

In this meta-analysis, we aimed to estimate the diagnostic accuracy of machine learning models on digital mammograms and tomosynthesis in breast cancer classification and to assess the factors affecting its diagnostic accuracy. We searched for related studies in Web of Science, Scopus, PubMed, Google Scholar and Embase. The studies were screened in two stages to exclude the unrelated studies and duplicates. Finally, 36 studies containing 68 machine learning models were included in this meta-analysis. The area under the curve (AUC), hierarchical summary receiver operating characteristics (HSROC) curve, pooled sensitivity and pooled specificity were estimated using a bivariate Reitsma model. Overall AUC, pooled sensitivity and pooled specificity were 0.90 (95% CI: 0.85–0.90), 0.83 (95% CI: 0.78–0.87) and 0.84 (95% CI: 0.81–0.87), respectively. Additionally, the three significant covariates identified in this study were country (*p* = 0.003), source (*p* = 0.002) and classifier (*p* = 0.016). The type of data covariate was not statistically significant (*p* = 0.121). Additionally, Deeks’ linear regression test indicated that there exists a publication bias in the included studies (*p* = 0.002). Thus, the results should be interpreted with caution.

## 1. Introduction

Breast cancer is the most commonly diagnosed cancer overall and among women worldwide; in fact, it has been identified as the fifth leading cause of cancer-related mortality globally in 2020 [1]. It is considered the most prevalent cancer worldwide [2]. The screening and diagnosis of breast cancer are carried out using multiple assessments, such as breast examination, mammography and biopsy. Different imaging modalities, such as mammography, ultrasound (US), magnetic resonance imaging (MRI), histological images and infrared thermography, have been used in breast cancer detection. Mammography is more commonly used for breast cancer screening. For example, women aged 40 years old and above are recommended to undergo a mammographic screening [3,4]. Mammography mainly consists of a digital mammogram and digital breast tomosynthesis (DBT). The digital mammogram is more commonly used for breast cancer detection; however, it is found to be less effective in patients with dense breasts and less sensitive to small tumors (tumors with a volume of less than 1 mm [5]). On the other hand, DBT or the three-dimensional mammogram, which is a more advanced technology of mammography, overcomes these disadvantages. Overall, it provides higher diagnostic accuracy than the two-dimensional mammogram [6]. However, no significant difference was noted between these two technologies when used for screening purposes [7].

Machine learning is expected to improve the area of health care, especially in medical specializations, such as diagnostic radiology, cardiology, ophthalmology and pathology [8]. Factors such as the availability of big medical data and advances in computing technology will help accelerate the use of machine learning in these medical areas. However, in spite of these positive developments, the practical implementation of machine learning in a clinical setting remains debatable [9,10,11]. Issues such as privacy concerns, lack of trust in the technology, machine learning interpretability and unintended bias of the technology are yet to be fully explored [8,12,13,14]. Machine learning had been researched to be used in the field of breast cancer in various ways, such as predicting and screening the disease [15], predicting the cancer recurrence [16], predicting survival of the patients [17], predicting the breast density and guiding treatments and management of the disease [18,19]. Different data sources, such as sociodemographic and clinical data, genomic data and imaging data, coupled with various machine learning techniques have been explored to be used in various clinical settings related to breast cancer. Thus, in brief, the use of machine learning in this research area can be categorized mainly into three roles, either as a screening, diagnostic or prognostic tool. These different roles of machine learning will affect how the model is built and deployed; however, most studies do not clearly emphasize the role of their machine learning model with regard to the clinical context and its practical application.

The use of machine learning on digital mammograms and tomosynthesis mainly aims to be a screening tool or at most, a supplemental diagnostic tool to a radiologist. Previous studies of machine learning on medical images associated with breast cancer mostly used digital mammograms [20], while the use of tomosynthesis was not very common. A wide variety of machine learning techniques has been used on these medical images, resulting in a wide range of diagnostic accuracy. Thus, the performance difference in all the techniques makes it difficult to evaluate the benefit of these machine learning tools on mammography. Subsequently, the wide range of performance of the machine learning techniques may reduce the confidence of the clinicians in the tools. Therefore, this meta-analysis aims to establish the overall diagnostic accuracy of the machine learning model on digital mammograms and tomosynthesis. This study also aims to assess the factors affecting the diagnostic accuracy of the machine learning model and further perform subgroup analysis.

## 2. Materials and Methods

### 2.1. Overview

This study was conducted according to the Preferred Reporting Items for Systematic Reviews and Meta-Analyses of diagnostic test accuracy studies (PRISMA-DTA) [21] and Synthesising Evidence from Diagnostic Accuracy Tests (SEDATE) [22] guidelines and recommendations. Both checklists are presented in the Appendix A.

### 2.2. Search Strategy

We searched the online databases of Scopus, PubMed, Google Scholar, Embase and Web of Science using predetermined search terms. The search was carried out on 17 August 2020 for Scopus, PubMed and Google Scholar databases. The search for Embase and Web of Science databases was conducted on 25 August 2020. All search terms for each database are presented in Appendix A.

All the results were imported into Mendeley. Duplicate papers were automatically screened and deleted. Subsequently, a researcher (TMH) manually screened the results again and deleted the remaining duplicates that were not identified using Mendeley. We then divided the screening process into two phases. In the first phase, we applied more lenient selection criteria to screen out the more obvious articles that were not related to our study. A full text of all the articles that passed the first phase of the selection criteria was downloaded. Additionally, in the second phase, we applied more stringent selection criteria to the articles to fit our study’s objectives. Any inconsistency during the selection and extraction process was resolved by discussion and consensus among the researchers.

### 2.3. Selection Criteria

We divided the screening process into two phases. We mainly screened the titles and abstracts and, if needed, the full text in the first phase. We searched for the following groups of articles in the first phase: (1) articles related to breast cancer prediction or classification; (2) articles that used machine learning models or algorithms; (3) articles written in English; (4) articles that used digital mammogram or tomosynthesis data; and (5) articles that at least reported an accuracy value as a performance metrics; (6) peer-reviewed research articles, proceedings and theses were excluded.

We screened all the articles using the full text in the second phase of the selection process. We selected the articles based on the following criteria: (1) articles that focused only on breast cancer classification models. Articles that compared feature extraction and segmentation methods were excluded. (2) Articles that reported a confusion matrix or at least had reported sufficient data. (3) Articles that had ensembles or hybrid machine learning models as classifiers were excluded. (4) Three-class prediction models were excluded unless a 2 × 2 confusion matrix was reported.

### 2.4. Data Extraction

We collectively extracted data from the included articles into a Microsoft Excel spreadsheet. The extracted variables were as follows: (1) title; (2) first author’s last name; (3) year of publication; (4) source of data; (5) country of the data used; (6) size of dataset; (7) number of data in the training, validation and testing split; (8) type of data; (9) sample size used; (10) classifier; (11) prediction class; (12) accuracy; (13) sensitivity; (14) specificity; and (15) confusion matrix. Additionally, more than one model was extracted from an article if the models used different data, classifiers or prediction classes. However, the model with the highest accuracy was extracted in the case of articles with relatively similar models.

### 2.5. Quality Assessment

We used the QUADAS-2 [23] tool to assess the quality of the studies that were included in the meta-analysis. The tool consisted of four domains, that is, patient selection, index test, reference standard, and flow and timing. All four domains were assessed regarding the risk of bias and only the first three domains were assessed regarding the applicability concerns. The risk of bias for each domain was determined using the signalling questions as entailed in the QUADAS-2 tool. Each signalling question was rated as ‘no’, ‘unclear’ or ‘yes’. The domains were considered a low risk of bias if all the signalling questions were rated ‘yes’. However, the domains were considered at a high risk of bias if one of the signalling questions was rated ‘no’ and none of the remaining signalling questions were rated ‘yes’. The domains, except for the previous two conditions, were considered an unclear risk of bias. Additionally, we added the overall rating to the QUADAS-2 assessment. We assigned the values of 1, 0 and −1, to low, unclear and high, respectively. Thus, the sum of the overall rating could range from −7 to 7. The overall quality was classified as very poor (−7 to −4), poor (−3 to 0), moderate (1 to 4) and good (5 to 7).

### 2.6. Outcomes

The primary outcomes were the overall diagnostic accuracy of the machine learning model in the form of the AUC and the hierarchical summary receiver operating characteristics (HSROC) curve. The secondary outcomes were the result of a likelihood ratio test for variables’ classifier, country of the data, source of data and type of the data. Variables with a *p*-value < 0.05 were considered statistically significant and followed up by a post hoc subgroup analysis.

### 2.7. Statistical Analysis

The statistical analysis was carried out using R version 4.1.0 [24]. The full R code is available on the GitHub website [25]. The main R packages used were *mada* and *metafor* [26,27]. A continuity correction of 0.5 was applied to the data if there were zero cells in the confusion matrix to avoid statistical artefacts. This approach is the default setting in the *mada* package. Each machine learning model was summarized by the pooled diagnostic odds ratio (DOR), sensitivity and specificity. The DOR represents the odds of a positive test result in diseased individuals compared to the odds of a positive result in healthy individuals. Thus, the DOR simply denotes the discriminant ability of the diagnostic test. Additionally, sensitivity represents the ability of the test to correctly identify affected individuals, while specificity reflects the ability of the test to correctly identify healthy individuals among the tested individuals. The pooled sensitivity, pooled specificity, AUC and HSROC curve parameters were estimated using the bivariate model of Reitsma et al. [28] through the *mada* package. The bivariate approach provides a better estimate, especially if a different cut-off threshold was used by each machine learning model to classify the positive and negative cases [22]. The 95% confidence interval of the AUC was estimated using a bootstrap method from the *dmetatools* package [29]. Heterogeneity assessment was conducted through visual inspection of the HSROC plot and the correlation between sensitivity and specificity. Inconsistency was suspected if the individual studies largely deviated from the HSROC line and the coefficient correlation of sensitivity and specificity was larger than zero [22,30]. The Cochran’s Q test and Higgins’ *I*^2^ statistics were not presented, as they were not suitable for heterogeneity assessments in diagnostic test accuracy studies [31].

A likelihood ratio test between the bivariate meta-regression models was carried out to compare a null model and a model with a covariate. Five bivariate meta-regression models were built, including the null model and models with a covariate of country, source, type of data and classifier. The country covariate indicated the country of origin of the data, while source covariate indicated whether the data were from a local database (primary data) or an online secondary database. The type of data covariate reflected the type of mammogram image and the classifier covariate reflected the different machine learning models included in this study. The likelihood ratio test with a *p*-value < 0.05 indicated that the model with a variable was better; thus, the variable was statistically significant. Subsequently, a post hoc subgroup analysis was performed for each significant variable. Pairwise comparisons of the AUC between each model of the subgroups were performed using a bootstrap method in the *dmetatools* package, and *p*-values were adjusted using the Bonferroni correction. A *p*-value below a threshold of 0.05 divided by the number of groups in each subgroup analysis indicated a significant comparison. A non-convergent result indicated that the model did not converge, even after 10,000 bootstrap resampling. Any subgroup model with a small number of studies was dropped from the subgroup analysis, as the estimates of the AUC and HSROC parameters were not reliable.

An influential diagnostic analysis was performed to assess the overall diagnostic accuracy of the machine learning model using the *dmetatools* package. The influential diagnostic analysis was carried out using a leave-one-out approach to estimate the difference in the AUC. Publication bias was evaluated using Deeks’ regression test [32]. The approach of Deeks et al., had been considered the most appropriate one to assess the publication bias in a diagnostic test accuracy study [33]. *p*-values < 0.10 may indicate the presence of publication bias.

## 3. Results

### 3.1. Eligible Studies

In total, 2897 research articles were identified in the 5 databases, as presented in Figure 1. After the removal of 1115 duplicates, the remaining 1782 articles were included in the screening process. A total of 1346 articles were excluded during the whole screening process. The first screening process excluded 1157 articles, while the second screening process excluded another 189 papers. Finally, 36 studies containing 68 machine learning models were included in this study.

### 3.2. Study Characteristics

The main characteristics of the included studies are presented in Table 1. The years of publication of the 36 included studies ranged from 2006 to 2020. Eleven studies used primary data from their respective countries, while most studies used secondary databases, such as the Mammographic Image Analysis Society (MIAS), mini-MIAS and Digital Database for Screening Mammography (DDSM). Only one study used tomosynthesis images, while the remaining thirty-five used digital mammogram images. The three most common classifiers were neural network (23.5%), support vector machine (22.1%) and deep learning (20.6%).

### 3.3. Descriptive Statistics

The study with the highest accuracy was the study carried out by Acharya U et al., in 2008 (98.3%), while that performed by Kanchanamani et al., in 2016 had the lowest accuracy (48.3%). The specificity and sensitivity values of each machine learning model are presented in Figure 2. Sensitivity values for machine learning models in this study ranged between 0.03 (95% CI: 0.00–0.24) and 1.00 (95% CI: 0.98–1.00), while specificity values ranged between 0.37 (95% CI: 0.25–0.50) and 0.98 (95% CI: 0.93–1.00). In this study, significant differences were observed between the sensitivity values (*p* < 0.001) and specificity values (*p* < 0.001) of machine learning models. The pooled DOR of the machine learning models was 28.34 (95% CI: 17.67–45.45), with the DOR value of each model ranging from 0.90 (95% CI: 0.44–1.84) to 7513.55 (95% CI: 445.61–126,689.03). Figure 3 presents the DOR values for each machine learning model in this study.

### 3.4. Overall Model

The pooled area under the curve (AUC) estimated using the bivariate model of Reitsma et al. [28] for the overall machine learning models in this study was 0.90 (95% CI: 0.85–0.90). The HSROC curve plot is presented in Figure 4. Additionally, the pooled sensitivity and pooled specificity values estimated through the same model were 0.83 (95% CI: 0.78–0.87) and 0.84 (95% CI: 0.81–0.87), respectively.

### 3.5. Test for Heterogeneity and Influential Diagnostics

Based on the HSROC curve plot (Figure 4), there was a moderate deviation of the individual models from the curve. The correlation coefficient of the sensitivity and specificity was 0.33. Thus, there was an indication of slight-to-moderate heterogeneity in this study. However, the influential diagnostics indicated that there was no influential model in the study. The result of the influential diagnostics is presented in Appendix A.

### 3.6. Subgroup Analysis

As per our findings, three out of four covariates were found to be significant via a likelihood ratio test; these were country (*p* = 0.003), source (*p* = 0.002) and classifier (*p* = 0.016), while the type of data was not significant (*p* = 0.121). The detailed result of the likelihood test is presented in Table 2. Thus, the country, source and classifier explained some of the heterogeneity that can be observed in the study. A further subgroup analysis was performed on the three significant covariates. All countries other than the USA and the UK were combined into one group, due to the small number of available studies. Subsequently, the studies that used data from both the USA and UK were excluded due to a small number of available studies, and those studies did not fit into any other group. Pairwise post hoc comparison of the country subgroup revealed that machine learning models that used data from the USA performed better than models that used data from the other countries in terms of AUC (dAUC = 0.10, 95% CI: 0.04–0.19). Additionally, for the subgroup analysis of the classifier covariate, three classifiers that were dropped due to a small number of studies were the Gaussian mixture model (GMM), linear discriminant analysis (LDA) and logistic regression. The three significant pairwise comparisons for this subgroup analysis were the neural network and Bayes-based model (dAUC = 0.25, 95% CI: 0.12–0.38), tree-based model and Bayes-based model (dAUC = 0.25, 95% CI: 0.07–0.40) and support vector machine and Bayes-based model (dAUC = 0.22, 95% CI: 0.09–0.35). Lastly, for the subgroup analysis of the source covariate, we dropped studies that used the INbreast database and the mammographic mass database (MMD). We also dropped studies that used both DDSM and MIAS databases and studies with unknown sources of data. Studies that used the MIAS and mini-MIAS databases were further classified into a single group. All pairwise comparisons of the AUC were determined to be not significant in this subgroup analysis. All the aforementioned pairwise comparisons were significant after the Bonferroni correction, and there were six non-convergent pairwise comparisons. The results of the complete pairwise comparisons for all the three subgroups are presented in Table 3, while Figure 5 delineates the HSROC for the subgroups. The highest AUCs in each subgroup were models with the US data (AUC = 0.94), models that used the DDSM database (AUC = 0.97) and the neural network model (0.94). As shown in Figure 5, models that used the DDSM database performed significantly better than models that used primary data, while the other model comparisons were relatively similar to those in Table 3.

### 3.7. Publication Bias

Deeks’ regression test was performed on the overall models that included all the 68 models from the 36 studies. The test indicated the possibility of publication bias in this study (*p* = 0.002). Figure 6 shows that Deeks’ funnel plot was asymmetrical.

### 3.8. Quality Assessment

Table 4 shows the quality assessment of the 36 included studies using the updated Quality Assessment of Diagnostic Accuracy Studies (QUADAS-2) tool. Generally, the majority of studies had an unclear risk of bias and low applicability concerns. Additionally, several studies with a high risk of bias were observed under the subdomains of ‘patient selection’ and ‘flow and timing’ of the risk of bias domain. Most studies used secondary databases and did not explain in detail the data selection process and flow of their studies. Items such as the consecutive or random sampling approach, inappropriate exclusion of the data and the proper interval between the index test and the reference standard were not clearly addressed in most of the included studies. Overall, out of the 36 studies included in the meta-analysis, 2 studies were found to be of poor quality, 9 studies of good quality and 25 studies of moderate quality.

## 4. Discussion

This study presents the efficacy of machine learning models on digital mammograms and tomosynthesis. According to our findings, machine learning models had good performance in breast cancer classification using digital mammograms and tomosynthesis, with pooled AUC of 0.90. A previous meta-analysis that analyzed different machine learning algorithms to estimate breast cancer risk was published in 2018 [70]. However, this study did not include deep learning methods and presented a summarized result for the overall machine learning methods. Another meta-analysis study focusing on deep learning reported good diagnostic accuracy for breast cancer detection using a mammogram, US, MRI and DBT with pooled AUCs of 0.87, 0.91, 0.87 and 0.91, respectively [71]. However, several meta-analysis studies that assessed the diagnostic accuracy of machine learning models on MRI in gliomas, prostate cancer and meningioma reported slightly lower AUCs of 0.88, 0.86 and 0.75, respectively [72,73,74]. This study included all previous studies that used any machine learning algorithms on mammography for breast cancer detection. In brief, the findings of our study support the promising potential use of machine learning on mammographic data for breast cancer detection in clinical settings, especially as a screening tool and a supplementary diagnostic tool to a radiologist.

Inconsistency among the diagnostic accuracy studies is to be expected [22]. In this meta-analysis, the three covariates that may explain the inconsistency among the studies were country, source and classifier. In terms of country, studies that used data from the USA and the UK had higher AUCs compared to the other countries (others group); however, only a pairwise comparison of the USA and other countries revealed a statistically significant result. This significant result may indicate a difference in characteristics between patients with breast cancer across countries. For example, breast cancer presentation and breast density had been reported to vary across populations [75,76], which, in turn, could affect the diagnostic accuracy of machine learning models. Additionally, this study found that studies that used primary data had lower AUCs compared to studies that used secondary databases. The studies that used primary data may reflect the actual diagnostic accuracy of machine models in real practice, as the data were collected specifically for the studies in question. Lastly, this study found that the classifier with the best AUC was the neural network, followed by the tree-based classifier and deep learning. However, the confidence regions of all these three models overlapped with each other (Figure 5), which indicated that none of the machine learning models significantly outperformed the other in terms of breast cancer classification. It is worth noting that one of the findings of this study was that the Bayes-based machine learning model had the lowest AUC (0.69) and performed significantly worse than the neural network, tree-based model and support vector machine. Nevertheless, a few studies were dropped in each subgroup analysis due to a small number of studies in that particular group, which limited the pairwise comparison that could be performed in each subgroup analysis. In brief, the subgroup analysis in this study showed that most machine learning models, such as the neural network (AUC = 0.938), deep learning (AUC = 0.918), tree-based models (0.934) and SVM (AUC = 0.904), perform well with mammographic data for breast cancer detection. Additionally, future studies should note that the characteristics and the quality of the mammographic data influence the performance of machine learning for breast cancer detection.

Despite the good performance of machine learning on mammography to be utilized for breast cancer detection, several considerations should be noted. Only 31% of the studies included in this meta-analysis used primary data collected by the researchers themselves, while the remaining 69% of the studies used publicly available datasets, such as MIAS, mini-MIAS and DDSM. Thus, future studies should focus on using high-quality data collected from the hospitals or research centers with a wide range of women with varying clinical symptoms of breast cancer. Furthermore, future studies should explicitly elucidate the role of machine learning tools that they develop either as screening, diagnostic or prognostic tools. Different roles of machine learning tools have different clinical impacts in the implementation of the tools. For example, machine learning screening tools should aim to reduce false-negative cases. Misdiagnosing a case with a high probability of breast cancer to a normal case is a fatal error. However, machine learning diagnostic tools should aim to reduce false-positive cases. Misdiagnosing a normal case as a breast cancer case will lead to unnecessary procedures, especially if it is an invasive procedure, such as a biopsy. Being transparent about where the machine learning tools can be implemented in the context of the clinical pathway of the disease increases the confidence of the clinicians in its utilization in the clinical setting. Nonetheless, there are many opportunities and benefits for the implementation of machine learning in breast cancer detection using mammographic data. The utilization of machine learning in breast cancer detection will reduce the workload of clinicians and accelerate the diagnosis workflow of the disease. Thus, breast cancer patients will receive early treatment, which further reduces the mortality rate of the disease.

In this study, we established the good performance of machine learning models on mammography in the classification of breast cancer. We used the bivariate model to estimate the AUC and further applied a bootstrap method to estimate its confidence interval. Furthermore, our meta-analysis included a reasonable number of studies to provide a relatively reliable result on the primary outcome and secondary outcomes. However, our study had several limitations. Firstly, we found that our study had a potential publication bias. One of the probable causes was the unpublished studies with a low-performance model. Additionally, the overall model in this study had a moderate amount of heterogeneity, and this study included a considerable number of studies that may contribute to both the occurrence of publication bias and the high statistical power of the asymmetry test. As shown in Figure 6, model 10 had a much higher DOR compared to the other models on the right side of the figure; however, removing this model did not have a significant impact on the AUC (Appendix A). Nonetheless, the mechanism of publication bias in diagnostic accuracy studies remains unclear, and a robust assessment of this bias is yet to be proposed [33]. Future meta-analyses may consider including the preprint articles that may be able to reduce the publication bias. Secondly, we only had one study with tomosynthesis, while the rest of the studies used digital mammograms. Thus, the findings of our study were more inclined toward digital mammograms than tomosynthesis, although both are considered mammography technology. In addition, we limited the language of the included studies to English, which may have increased the risk of bias in our findings. Lastly, there are a wide variety of machine learning models with different variants and parameters available. Thus, our study was not able to compare each of the model variants, due to the lack of sample size of that particular model.

## 5. Conclusions

In conclusion, the performance of machine learning on mammography in breast cancer classification showed promising results, with good sensitivity and specificity values. However, the role of any machine learning technique in the diagnostic pathway should be clearly explained in a diagnostic accuracy study to be efficiently incorporated into the clinical setting. Thus, the limitation of each machine learning model will be apparent to clinicians and other health personnel.

## Figures and Tables

**Figure 1 diagnostics-12-01643-f001:**
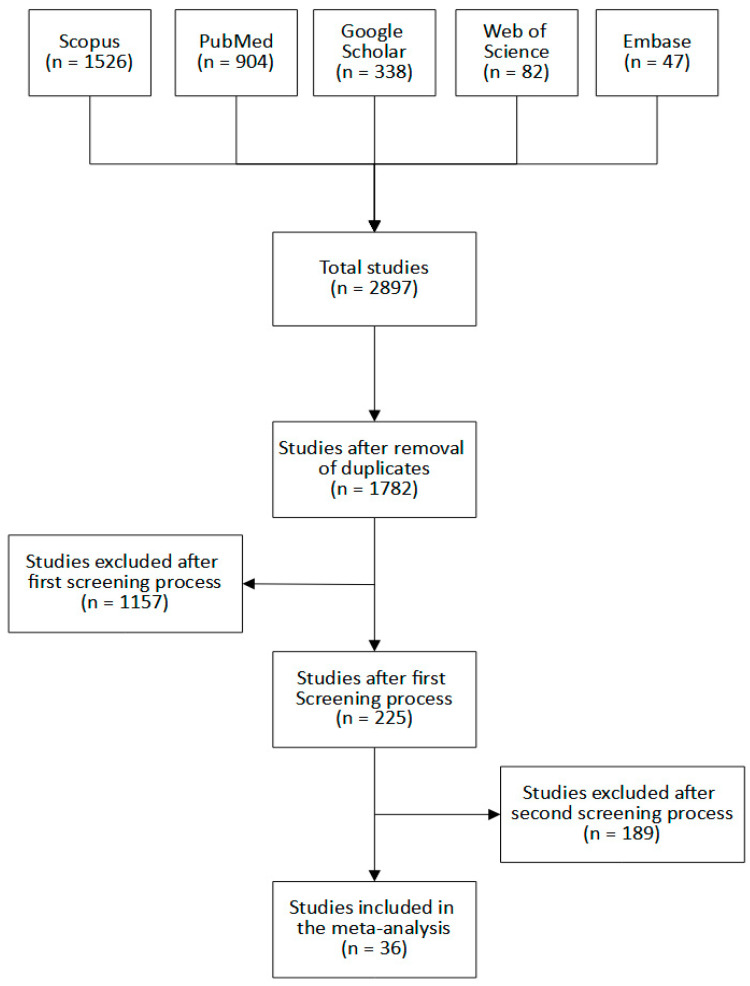
Flow diagram of the study selection process.

**Figure 2 diagnostics-12-01643-f002:**
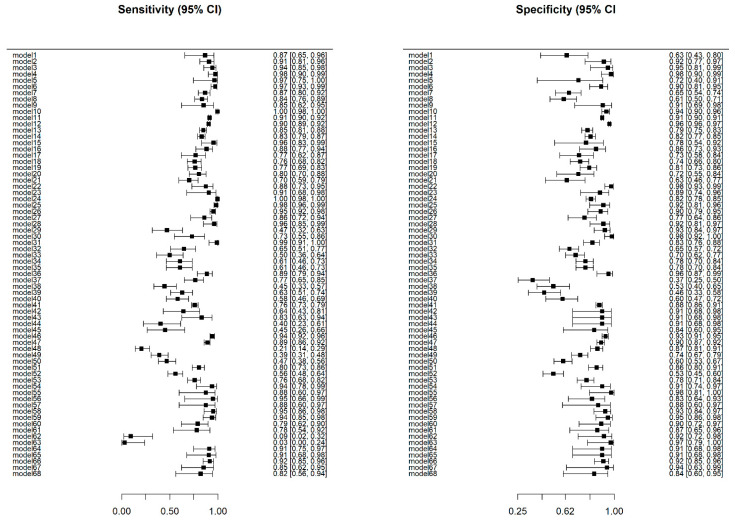
Sensitivity and specificity of machine learning models in the study.

**Figure 3 diagnostics-12-01643-f003:**
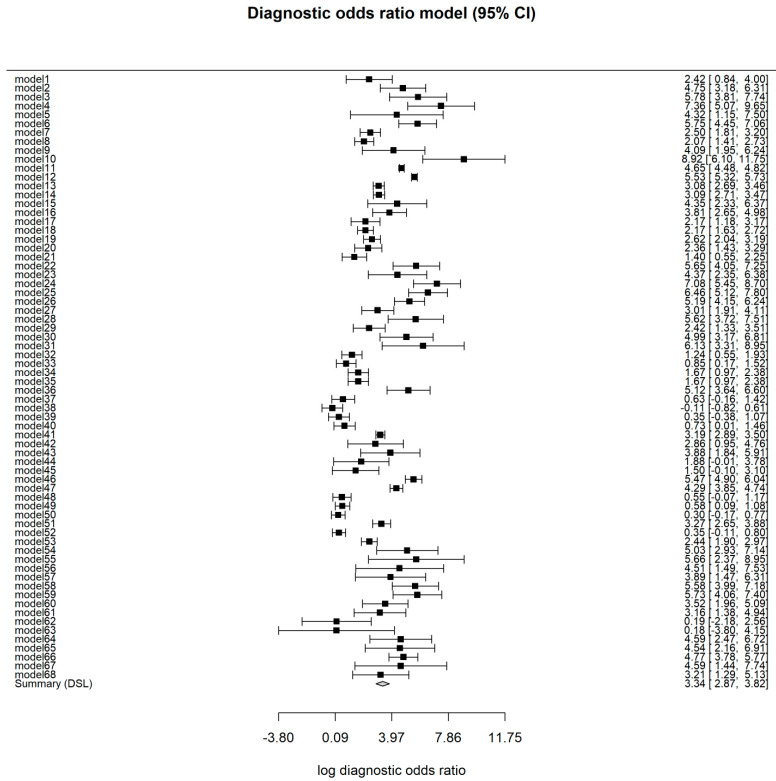
The diagnostic odds ratio of machine learning models in the study.

**Figure 4 diagnostics-12-01643-f004:**
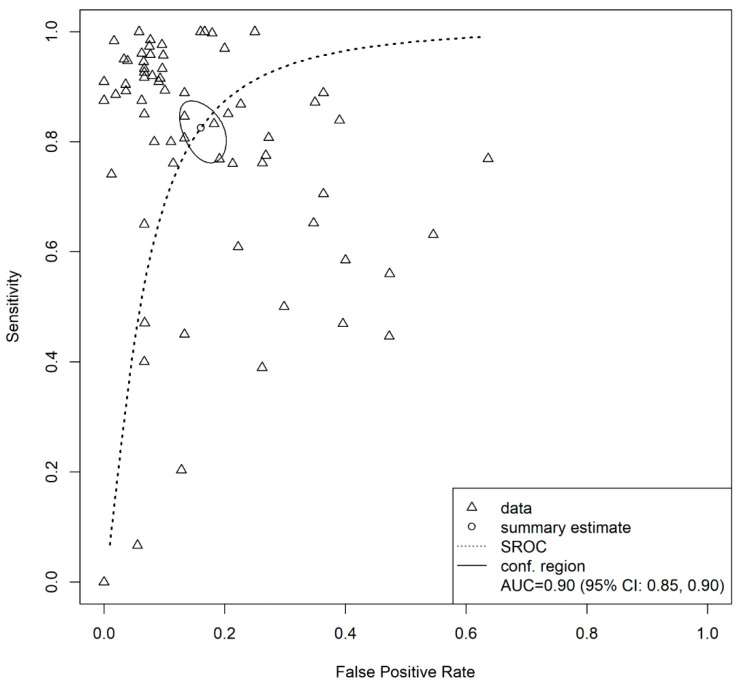
Hierarchical summary receiver operating characteristics (HSROC) curve for overall machine learning models in the study.

**Figure 5 diagnostics-12-01643-f005:**
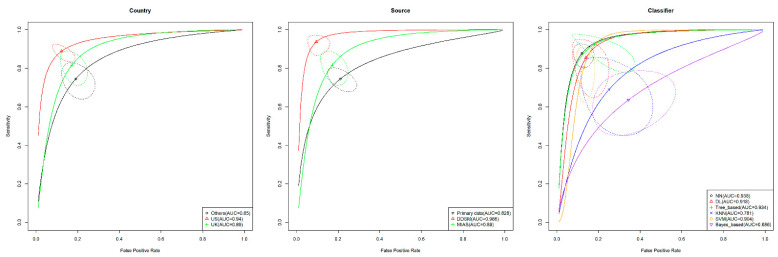
Hierarchical summary receiver operating characteristics (HSROC) curve for each subgroup analysis in the study.

**Figure 6 diagnostics-12-01643-f006:**
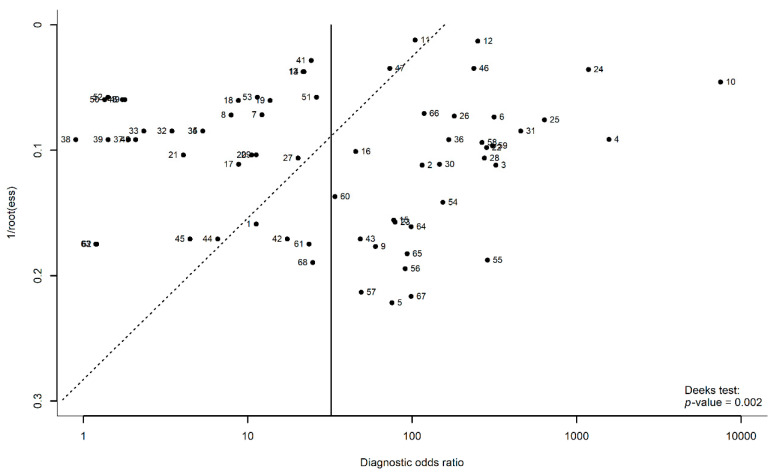
Deeks’ funnel plot.

**Table 1 diagnostics-12-01643-t001:** Characteristics of included studies.

Study	ID	Country	Source	Size of Dataset	Train/Validation/Test Split	Type of Data	Classifier	Prediction Class	TP	TN	FP	FN	Accuracy
Abdolmaleki 2006 [34]	1	Iran	Primary data	122 cases	82/-/40	DM	NN	Benign-Malignant	16	14	8	2	0.75
Acharyau 2008 [35]	2	USA	DDSM	360 images	270/-/90	DM	NN	Normal-Benign-Malignant	55	28	2	5	0.97
	3	USA	DDSM	360 images	270/-/90	DM	GMM	Normal-Benign-Malignant	57	29	1	3	0.98
Al-antari 2020 [36]	4	USA	DDSM	600 images	420/60/120	DM	DL	Benign-Malignant	59	59	1	1	0.98
	5	Portugal	INbreast	410 images	78/12/22	DM	DL	Benign-Malignant	14	6	2	0	0.95
Alfifi 2020 [37]	6	UK	MIAS	200 images	NE	DM	DL	Normal-Benign-Malignant	124	66	7	3	0.95
	7	UK	MIAS	200 images	NE	DM	Tree-based	Normal-Benign-Malignant	102	54	29	15	0.78
	8	UK	MIAS	200 images	NE	DM	KNN	Normal-Benign-Malignant	99	50	32	19	0.74
Al-hiary 2012 [38]	9	Jordan	Primary data	NE	NE	DM	NN	Normal-Cancer	14	15	1	2	0.91
Al-masni 2018 [39]	10	USA	DDSM	2400 images	1920/-/480	DM	NN	Benign-Malignant	240	226	14	0	0.97
Bandeira-diniz 2018 [40]	11	USA	DDSM	2482 images	1990/-/492	DM	DL	Non-mass-Mass	2418	4306	442	225	0.91
	12	USA	DDSM	2482 images	1990/-/492	DM	DL	Non-mass-Mass	1774	5615	210	188	0.95
Barkana 2017 [41]	13	USA	DDSM	2173 images	1451/-/722	DM	NN	Benign-Malignant	325	270	70	57	0.82
	14	USA	DDSM	2173 images	1451/-/722	DM	SVM	Benign-Malignant	318	278	62	64	0.83
Biswas 2019 [42]	15	UK	MIAS	322 images	226/48/48	DM	NN	Normal-Abnormal	32	12	3	1	0.92
Cai 2019 [43]	16	China	Primary data	990 images	891/-/99	DM	SVM	Benign-Malignant	48	39	6	6	0.89
Chen 2019a [44]	17	China	Primary data	81 cases	NE	DM	Tree-based	Benign-Malignant	31	30	11	9	0.75
Chen 2019b [45]	18	USA	Primary data	275 cases	10-folds cross validation	DM	SVM	Benign-Malignant	102	104	37	32	0.75
	19	USA	Primary data	275 cases	10-folds cross validation	DM	SVM	Benign-Malignant	103	114	27	31	0.79
Danala 2018 [46]	20	USA	Primary data	111 cases	LOO-CV	DM	DL	Benign-Malignant	63	24	9	15	0.78
	21	USA	Primary data	111 cases	LOO-CV	DM	DL	Benign-Malignant	55	21	12	23	0.68
Daniellopez-cabrera 2020 [47]	22	UK	mini-MIAS	322 images	NE	DM	DL	Normal-Abnormal	31	101	2	4	0.97
	23	UK	mini-MIAS	322 images	NE	DM	DL	Benign-Malignant	14	28	3	1	0.91
Fathy 2019 [48]	24	USA	DDSM	3932 images	2517/629/786	DM	DL	Normal-Abnormal	389	325	71	1	0.91
Girija 2019 [49]	25	UK	mini-MIAS	322 images	NE	DM	Tree-based	Normal-Abnormal	266	48	4	4	0.98
	26	UK	mini-MIAS	322 images	NE	DM	Tree-based	Benign-Malignant	200	55	6	9	0.94
Jebamony 2020 [50]	27	UK	mini-MIAS	294 images	203/-/91	DM	NN	Benign-Malignant	33	41	12	5	0.85
	28	UK	mini-MIAS	294 images	203/-/91	DM	SVM	Benign-Malignant	37	49	4	1	0.96
Junior 2010 [51]	29	UK	mini-MIAS	428 ROIs	320/-/108	DM	NN	Normal-Abnormal	16	69	5	18	0.79
	30	UK	mini-MIAS	428 ROIs	320/-/108	DM	SVM	Normal-Abnormal	20	80	1	7	0.93
Kanchanamani 2016 [52]	31	UK	MIAS	322 images	NE	DM	SVM	Normal-Abnormal	46	120	24	0	0.87
	32	UK	MIAS	322 images	NE	DM	Bayes-based	Normal-Abnormal	30	94	50	16	0.65
	33	UK	MIAS	322 images	NE	DM	DL	Normal-Abnormal	23	101	43	23	0.65
	34	UK	MIAS	322 images	NE	DM	KNN	Normal-Abnormal	28	112	32	18	0.74
	35	UK	MIAS	322 images	NE	DM	LDA	Normal-Abnormal	28	112	32	18	0.74
	36	UK	MIAS	322 images	NE	DM	SVM	Benign-Malignant	58	53	2	7	0.93
	37	UK	MIAS	322 images	NE	DM	Bayes-based	Benign-Malignant	50	20	35	15	0.58
	38	UK	MIAS	322 images	NE	DM	DL	Benign-Malignant	29	29	26	36	0.48
	39	UK	MIAS	322 images	NE	DM	KNN	Benign-Malignant	41	25	30	24	0.55
	40	UK	MIAS	322 images	NE	DM	LDA	Benign-Malignant	38	33	22	27	0.59
Kim 2018 [53]	41	Korea	Primary data	29,107 images	26631/1238/1238	DM	DL	Normal-Abnormal	471	548	71	148	0.82
Mao 2019 [54]	42	China	Primary data	173 cases	138/-/35	DM	SVM	Benign-Malignant	13	14	1	7	0.80
	43	China	Primary data	173 cases	138/-/35	DM	Logistic	Benign-Malignant	17	14	1	3	0.89
	44	China	Primary data	173 cases	138/-/35	DM	KNN	Benign-Malignant	8	14	1	12	0.83
	45	China	Primary data	173 cases	138/-/35	DM	Bayes-based	Benign-Malignant	9	13	2	11	0.78
Miao 2015 [55]	46	USA	MMD	830 cases	10-folds cross validation	DM	SVM	Benign-Malignant	381	399	28	22	0.94
Miao 2013 [56]	47	USA	MMD	830 cases	NE	DM	NN	Benign-Malignant	360	384	43	43	0.90
Milosevic 2015 [57]	48	UK	MIAS	300 images	5-folds cross validation	DM	SVM	Normal-Abnormal	23	163	24	90	0.62
	49	UK	MIAS	300 images	5-folds cross validation	DM	KNN	Normal-Abnormal	44	138	49	69	0.61
	50	UK	MIAS	300 images	5-folds cross validation	DM	Bayes-based	Normal-Abnormal	53	113	74	60	0.55
	51	Serbia	Primary data	300 images	5-folds cross validation	DM	SVM	Normal-Abnormal	121	130	20	29	0.84
	52	Serbia	Primary data	300 images	5-folds cross validation	DM	KNN	Normal-Abnormal	84	79	71	66	0.54
	53	Serbia	Primary data	300 images	5-folds cross validation	DM	Bayes-based	Normal-Abnormal	114	118	32	36	0.77
Nithya 2012 [58]	54	USA	DDSM	250 images	200/-/50	DM	NN	Normal-Abnormal	23	24	2	1	0.94
Nusantara 2016 [59]	55	UK	MIAS	322 images	291/-/31	DM	KNN	Normal-Abnormal	10	20	0	1	0.97
Palantei 2017 [60]	56	UK	MIAS	NE	NE	DM	SVM	Normal-Abnormal	9	21	4	0	0.88
Paramkusham 2018 [61]	57	USA	DDSM	148 images	126/-/22	DM	SVM	Benign-Malignant	10	10	1	1	0.91
Roseline 2018 [62]	58	UK	MIAS	NE	NE	DM	KNN	Benign-Malignant	49	60	4	2	0.95
Shah 2015 [63]	59	UK	MIAS	320 images	NE	DM	NN	Normal-Abnormal	54	49	2	3	0.95
	60	UK	MIAS	320 images	NE	DM	NN	Benign-Malignant	24	22	2	6	0.85
Shivhare 2020 [64]	61	USA, UK	DDSM, MIAS	NE	NE	DM	NN	Benign-Malignant	12	16	2	3	0.85
	62	USA, UK	DDSM, MIAS	NE	NE	DM	DL	Benign-Malignant	1	17	1	14	0.55
	63	USA, UK	DDSM, MIAS	NE	NE	DM	SVM	Benign-Malignant	0	18	0	15	0.55
Singh 2018 [65]	64	UK	MIAS	139 ROIs	69/28/42	DM	NN	Benign-Malignant	25	14	1	2	0.93
Venkata 2019 [66]	65	NA	NA	110 images	80/-/30	DM	Logistic regression	Benign-Malignant	14	14	1	1	0.93
Wang 2017 [67]	66	UK	mini-MIAS	200 images	10-folds cross validation	DM	NN	Normal-Abnormal	92	92	8	8	0.92
Wutsqa 2017 [68]	67	UK	MIAS	120 cases	96/-/24	DM	NN	Normal-Abnormal	14	8	0	2	0.92
Yousefi 2018 [69]	68	USA	Primary data	87 images	NE	Tomosynthesis	Tree-based	Benign-Malignant	11	13	2	2	0.87

DM = digital mammogram; NN = neural network; GMM = Gaussian mixture model; DL = deep learning; KNN = k-nearest neighbor; SVM = support vector machine; LDA = linear discriminant analysis; ROIs = region of interests; LOO-CV = leave-one-out cross validation; NE = not clearly explained; NA = not available; TP = true positive; TN = true negative; FP = false positive; FN = false negative; DDSM = database for screening mammography; MIAS = mammographic image analysis society; MMD = mammographic mass database.

**Table 2 diagnostics-12-01643-t002:** A likelihood ratio test for bivariate meta-regression models with the null model.

Model	Covariate	ꭓ2-Statistic (df)	*p*-Value
Model 1	Country	19.55 (6)	0.003 *
Model 2	Source	31.10 (12)	0.002 *
Model 3	Type of data	4.23 (2)	0.121
Model 4	Classifier	30.32 (16)	0.016 *

* Significance at *p* < 0.05.

**Table 3 diagnostics-12-01643-t003:** A post hoc pairwise comparison for covariates country, source of data and classifier.

Comparisons	dAUC (95% CI)	*p*-Value
Country		
USA vs. UK	0.051 (0.006, 0.127)	0.035 *
USA vs. others ^1^	0.095 (0.044, 0.191)	0.001 **
UK vs. others ^1^	0.044 (−0.034, 0.131)	0.241
Source of data		
Primary data vs. DDSM	— ^†^	— ^†^
Primary data vs. MIAS ^2^	−0.062 (−0.127, 0.023)	0.152
DDSM vs. MIAS ^2^	— ^†^	— ^†^
Classifier		
NN vs. DL	— ^†^	— ^†^
NN vs. Tree-based	0.003 (−0.071, 0.138)	0.946
NN vs. KNN	0.157 (0.026, 0.325)	0.010
NN vs. SVM	0.033 (−0.034, 0.074)	0.337
NN vs. Bayes-based	0.252 (0.119, 0.379)	<0.001 **
DL vs. Tree-based	−0.016 (−0.122, 0.117)	0.690
DL vs. KNN	— ^†^	— ^†^
DL vs. SVM	— ^†^	— ^†^
DL vs. Bayes-based	— ^†^	— ^†^
Tree-based vs. KNN	0.153 (−0.023, 0.333)	0.082
Tree-based vs. SVM	0.030 (−0.101, 0.099)	0.578
Tree-based vs. Bayes-based	0.249 (0.073, 0.395)	0.007 **
KNN vs. SVM	−0.123 (−0.300, −0.004)	0.044 *
KNN vs. Bayes-based	0.096 (−0.121, 0.265)	0.404
SVM vs. Bayes-based	0.219 (0.094, 0.350)	<0.001 **

* Significance at *p* < 0.05; ** significance after Bonferroni correction; ^†^ non-convergence; ^1^ others: Iran, Portugal, Jordan, China, Korea and Serbia; ^2^ mini-MIAS and MIAS databases were combined into a group; dAUC = difference of the area under the curve; DDSM = database for screening mammography; MIAS = mammographic image analysis society; NN = neural network; DL = deep learning; KNN = k-nearest neighbor; SVM = support vector machine.

**Table 4 diagnostics-12-01643-t004:** Quality assessment of the included studies according to the QUADAS-2 tool.

**Study**	**Risk of** **Bias**				**Applicability**			**Overall**
	**Patient Selection**	**Index Test**	**Reference Standard**	**Flow and Timing**	**Patient Selection**	**Index** **Test**	**Reference Standard**	
Abdolmaleki 2006	Low	Unclear	Low	Low	Low	Low	Low	Good
Acharyau 2008	High	Unclear	Low	Unclear	Low	Low	Low	Good
Al-antari 2020	Low	Unclear	Unclear	Low	Unclear	Low	Unclear	Moderate
Alfifi 2020	Unclear	Unclear	Unclear	Unclear	Low	Low	Unclear	Moderate
Al-hiary 2012	High	Low	Unclear	Unclear	Unclear	Low	Unclear	Moderate
Al-masni 2018	Low	Unclear	Low	Unclear	Low	Low	Low	Moderate
Bandeira-diniz 2018	High	Low	Low	Unclear	Low	Low	Low	Good
Barkana 2017	Unclear	Unclear	Low	Unclear	Unclear	Low	Low	Moderate
Biswas 2019	Unclear	Unclear	Unclear	Unclear	Unclear	Low	Unclear	Moderate
Cai 2019	Low	Low	Low	Low	Low	Low	Low	Moderate
Chen 2019a	Low	Unclear	Low	Low	Low	Low	Low	Moderate
Chen 2019b	Low	Low	Low	Low	Low	Low	Low	Good
Danala 2018	Low	Low	Low	Low	Low	Low	Low	Good
Daniellopez-cabrera 2020	Unclear	Unclear	Unclear	Unclear	Low	Low	Unclear	Good
Fathy 2019	High	Low	Low	Unclear	Low	Low	Low	Poor
Girija 2019	Unclear	Low	Unclear	Unclear	Low	Low	Low	Good
Jebamony 2020	Unclear	Unclear	Unclear	High	Low	Low	Unclear	Moderate
Junior 2010	High	Unclear	Unclear	High	Low	Low	Unclear	Moderate
Kanchanamani 2016	Unclear	Unclear	Unclear	Unclear	Low	Low	Unclear	Moderate
Kim 2018	Unclear	Low	Low	Low	Low	Low	Low	Moderate
Mao 2019	Low	Unclear	Low	Low	Low	Low	Low	Moderate
Miao 2015	Unclear	Unclear	Unclear	High	Low	Low	Unclear	Moderate
Miao 2013	Low	Low	Unclear	High	Low	Low	Unclear	Moderate
Milosevic 2015	Low	Unclear	Unclear	Unclear	Low	Low	Unclear	Moderate
Nithya 2012	Unclear	Unclear	Low	Unclear	Low	Low	Low	Moderate
Nusantara 2016	Unclear	Low	Unclear	Unclear	Low	Low	Low	Moderate
Palantei 2017	High	Unclear	Unclear	Unclear	Low	Low	Unclear	Poor
Paramkusham 2018	Unclear	Unclear	Low	Unclear	Low	Low	Low	Moderate
Roseline 2018	Unclear	Unclear	Unclear	High	Low	Low	Unclear	Moderate
Shah 2015	Unclear	Unclear	Unclear	Unclear	Low	Low	Unclear	Good
Shivhare 2020	Unclear	Unclear	Unclear	High	Low	Low	Unclear	Good
Singh 2018	Unclear	Unclear	Low	Low	Low	Low	Low	Moderate
Venkata 2019	Unclear	Unclear	Unclear	Unclear	Unclear	Low	Unclear	Moderate
Wang 2017	High	Unclear	Unclear	Unclear	Low	Low	Unclear	Moderate
Wutsqa 2017	High	Unclear	Unclear	Unclear	Low	Low	Unclear	Moderate
Yousefi 2018	Unclear	Unclear	Low	Unclear	Low	Low	Low	Moderate

## Data Availability

All data generated or analysed during this study are included in this published article.

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
