# Peer review of "Diagnostic Accuracy of Machine Learning Models on Mammography in Breast Cancer Classification: A Meta-Analysis"

_diagnostics, 2022, doi:10.3390/diagnostics12071643_

Round 1

Reviewer 1 Report

General comment: The review sought to assess the diagnostic accuracy of machine learning models for mammography and DBT and the factors affecting their performance. The paper is topical and addressed the first aim, but there is no data in the paper to support the second aim of the review, which was to assess the factors affecting the diagnostic accuracy of these models. This information should be included, or the second aim should be removed. The discussion needs to highlight the new findings from this meta-analysis and their relevance to policy and practice.

Abstract: The study also sought to assess the factors affecting the diagnostic accuracy of machine learning models. the results should be added to the abstract to make it standalone

Introduction: Breast cancer is not the most prevalent cancer. Please revise

Line. 28-31: DBT is not the standard for screening as alluded in the sentence.

The meta-analysis needs a stronger rationale. The authors need to provide a rationale why the diagnostic accuracy of machine learning models should be examined.

Selection criteria: The presentation of inclusion and exclusion criteria should be revised to enhance clarity. The inclusion criteria should be listed first followed by the exclusion criteria

Quality assessment. Quadas-2 has four domains, not 2. The authors should state how patient selection, index test, reference standard, and flow and timing were assessed.

Results3.1 Eligible studies. The authors should include the reasons for exclusion of the articles.

Results: Very well presented; however, separating the results of DBT from DM studies would have added more value to the results.

Discussion: The authors have done a good job of summarising the results, comparing them with the literature, and highlighting the inconsistencies across studies and limitations of the review. A discussion around what the meta-analysis has added to the literature, the implications of these new findings, and their relevance to policy and practice would have done more justice to the paper.

Figures: fine

References: fine

Reviewer 2 Report

Summary: A meta-analysis to find the efficacy of mammography in detecting breast cancer. This study found that the machine learning on mammography in breast cancer classification has promising accuracy, with good sensitivity and specificity values. 

Comments: 

1. It is recommended the authors re-write the Introduction section to better summarize the current understandings on the role of machine learning of breast mammography in cancer diagnosis, the literature gap, and the importance of the study. Besides, general information on machine learning can be more summarized. 

2. Introduction: "This study also aims to assess the factors affecting the diagnostic accuracy of the machine learning model and further perform subgroup analysis." The aim of this meta-analysis is unclear. What does "breast cancer classification" mean in the main title? Was this MA applied to clarify mammography efficacy to detect malignant tumor from benign tumor? or to classify different malignant tumors? Does it include studies applying mammography as a screening tool in asymptomatic patients or a diagnostic tool in symptomatic individuals? The characteristics of patients included in the studies have not been mentioned. These crucial issues requires to be explained in the introduction and methods sections.

3. Methods-Search strategy: please mention the search period, instead of the sentence "The search was carried out in August 2020. ".

4. Please explain the full-term of acronyms for their first appearance in the manuscript-e.g. DOR.

5. Table 1: Please explain all the acronyms in the table footnote.

6. Discussion: Literature contains several other MAs with the similar objective. Please compare the results with their findings. 

Round 2

Reviewer 1 Report

The authors have addressed most of my comments except the one about adding a discussion around what the meta-analysis has added to the literature, the implications of these new findings, and their relevance to policy and practice.

Author Response

We thank the reviewer for the comments and suggestions

Note – page numbers and lines are based on “Simple Markup” in Track Changes in the word file.

Response to reviewer 1:

Point 1: The authors have addressed most of my comments except the one about adding a discussion around what the meta-analysis has added to the literature, the implications of these new findings, and their relevance to policy and practice.

Response 1:

  • We have added a paragraph addressing the implication and relevance of the findings to the policy and practice (page: 17, line: 368-389).
  • We have mentioned what this meta-analysis added to the literature (page: 16, line: 327-339, page: 16-17, line: 364-367).
  • We hope these points address your concerns. Thank you.

Reviewer 2 Report

Thank you for addressing all the comments. I have no more comments.

Author Response

Thank you for the comments and suggestions.